# Energy Metabolism and Intracellular pH Alteration in Neural Spheroids Carrying Down Syndrome

**DOI:** 10.3390/biomedicines9111741

**Published:** 2021-11-22

**Authors:** Alena Kashirina, Alena Gavrina, Emil Kryukov, Vadim Elagin, Yuliya Kolesova, Alexander Artyuhov, Ekaterina Momotyuk, Vepa Abdyyev, Natalia Meshcheryakova, Elena Zagaynova, Erdem Dashinimaev, Aleksandra Kashina

**Affiliations:** 1Institute of Experimental Oncology and Biomedical Technologies, Privolzhsky Research Medical University, 603005 Nizhny Novgorod, Russia; gavrina.alena@mail.ru (A.G.); rodrigobordzhia@gmail.com (E.K.); elagin.vadim@gmail.com (V.E.); ezagaynova@gmail.com (E.Z.); almele@yandex.ru (A.K.); 2Institute of Molecular Medicine, Sechenov First Moscow State Medical University, 119991 Moscow, Russia; vasilenko-yuliya@mail.ru; 3Koltzov Institute of Developmental Biology, Russian Academy of Sciences, 119334 Moscow, Russia; mailtovepa@gmail.com (V.A.); dashinimaev@gmail.com (E.D.); 4Center for Precision Genome Editing and Genetic Technologies for Biomedicine, Pirogov Russian National Research Medical University, Ostrovitianov Street, 117997 Moscow, Russia; alexanderartyuhov@gmail.com (A.A.); edm95r@rambler.ru (E.M.); time-off@mail.ru (N.M.); 5Institute of Biology and Biomedicine, Lobachevsky State University of Nizhni Novgorod, 603022 Nizhny Novgorod, Russia; 6Moscow Institute of Physics and Technology, 141701 Dolgoprudny, Russia

**Keywords:** iPSC, spheroids, Down syndrome, metabolism, NAD(P)H, pH, BCECF, SypHer-2, FLIM

## Abstract

Brain diseases including Down syndrome (DS/TS21) are known to be characterized by changes in cellular metabolism. To adequately assess such metabolic changes during pathological processes and to test drugs, methods are needed that allow monitoring of these changes in real time with minimally invasive effects. Thus, the aim of our work was to study the metabolic status and intracellular pH of spheroids carrying DS using fluorescence microscopy and FLIM. For metabolic analysis we measured the fluorescence intensities, fluorescence lifetimes and the contributions of the free and bound forms of NAD(P)H. For intracellular pH assay we measured the fluorescence intensities of SypHer-2 and BCECF. Data were processed with SPCImage and Fiji-ImageJ. We demonstrated the predominance of glycolysis in TS21 spheroids compared with normal karyotype (NK) spheroids. Assessment of the intracellular pH indicated a more alkaline intracellular pH in the TS21 spheroids compared to NK spheroids. Using fluorescence imaging, we performed a comprehensive comparative analysis of the metabolism and intracellular pH of TS21 spheroids and showed that fluorescence microscopy and FLIM make it possible to study living cells in 3D models in real time with minimally invasive effects.

## 1. Introduction

Induced pluripotent stem cells (iPSCs) are attractive objects for scientific research in various fields of cell biology, one of which is the study of human hereditary diseases. It has been shown that iPSCs from individuals with hereditary diseases can develop into the disease-associated cell types and that the derived cells reproduce the metabolic hallmarks of the respective diseases, for example, Down syndrome, Alzheimer’s disease, Parkinson’s disease many other disorders [1,2,3,4,5]. However, traditional two-dimensional cell cultures do not adequately reflect the microenvironment of such cells in vivo. To address this and better mimic the complex environment of human tissues, great numbers of protocols have been developed for the generation of 3D cell models from iPSCs. 3D models can help in the study of the tissues of patients with hereditary diseases (for example, Down syndrome and the hereditary form of Alzheimer’s disease) in vitro since the only alternatives are postmortem tissues and biological fluids. 

Down syndrome is a genetic disease caused by trisomy of chromosome 21 (TS21), which has various manifestations, primarily a violation of brain functioning [6]. Several studies using cells differentiated from TS21 human iPSCs have shown that these objects reflect the molecular and cellular phenotypes of primary cells obtained from patients with DS [7]. It is known that brain diseases are accompanied by changes in many cellular parameters, including their metabolic activity. Early research showed that TS21 negatively affects mitochondrial function [8], and it has been demonstrated that, in human fibroblasts carrying DS, TS21 disrupts the expression of genes involved in the mitochondrial pathways and reduces the ATP content and O_2_ consumption by cells [9]. Mitochondrial damage associated with genetic abnormalities directly affects the course of the oxidative phosphorylation (OXPHOS) process.

The OXPHOS system is the main energy provider powering the activity of mature neurons [10]. An adequate course of this process is fundamental for the development of the brain and the neurogenic process during prenatal development [11,12]. This biochemical pathway is essential for neuronal differentiation. Pathological mutations in genes associated with OXPHOS impair neuronal differentiation [13,14,15]. Moreover, as OXPHOS is a process that occurs in all cells of the body its deficiency can affect every organ and, therefore, be associated with many diseases. Hence, any strategy developed to improve OXPHOS function could provide the basis of a successful therapeutic approach to DS.

To adequately assess metabolic changes and to test drugs aimed at correcting cellular metabolism in 3D systems with disease cells, methods are needed that allow the monitoring of changes in real time with minimally invasive effects on the samples. Immunocytochemical staining [16,17], histochemical staining [18], colorimetric methods and biochemical methods [19] are usually used for assessment of the metabolic changes in 3D cultures of cells. However, many of these techniques are indirect, invasive and terminal. Currently, optical bioimaging methods are used to solve such problems. Among modern bioimaging methods, fluorescence microscopy and FLIM (microscopy with the detection of fluorescence lifetimes) open a unique opportunity for the label free, non-invasive and long-term investigation of metabolic activity [20]. Fluorescence microscopy allows estimation of the optical redox ratio, which is used as an indicator of the mitochondrial redox state associated with NAD, and correlates well with the redox potential of NAD^+^/NADH as determined by biochemical methods [21]. In particular, higher values of the redox ratio are known to indicate the prevalence of OXPHOS [22]. The FLIM method allows measurement of the fluorescence lifetime contributions of both exogenous and endogenous fluorophores [23]. From the fluorescence lifetime contributions of NAD(P)H, the relative amount of the free or enzyme-bound form of the cofactor in the cell can be estimated [24].

The switching of metabolic pathways is directly related to the concentration of protons (pH) inside the cell. This can therefore serve as another indicator of metabolic rebalancing. More alkaline pH values are known to promote glycolysis in cells [25]. The physiological level of intracellular pH is maintained in the range of 7.1–7.2. A sharp shift in pH to the acidic or alkaline side may be associated with human diseases including neurodegenerative ones such as Alzheimer’s disease [26]. Using optical methods, the pH can be assessed either by staining with fluorescent probes, or by using genetically encoded sensors. The advantages of fluorescent probes are their selectivity, sensitivity, quickness of use, and relatively low cost [27,28]. Genetically encoded pH sensors are minimally invasive tools for the selective targeting of different cellular compartments [29]. Thus, at present, the use of optical methods allows us to study the tissues of patients (in 3D models) with diseases dynamically and non-invasively. In addition, the study of such parameters as metabolism and pH can help propel us forward in the development of new therapeutic approaches in the treatment of neurodegenerative diseases. 

The aim of this work was to study the metabolic status and pH of TS21 spheroids based on both the fluorescence of the metabolic cofactors NAD(P)H and FAD and their fluorescence lifetimes using fluorescence microscopy and FLIM, combined with methods involving fluorescence microscopy of spheroids containing the pH sensitive probe—BCECF and the genetically encoded sensor SypHer-2.

## 2. Materials and Methods

### 2.1. Cell Culture and Differentiation

iPS-KYOU is a line of induced pluripotent human stem cells obtained in the Shinya Yamanaka laboratory (University of Kyoto, Kyoto, Japan) by lentiviral reprogramming using the OCT4, SOX2, c-MYC and KLF4 factors of adult skin fibroblasts. iPS-DYP0730 is a line of induced pluripotent human stem cells obtained by the episomal reprogramming of adult skin fibroblasts of a patient with Down syndrome. The iPS-KYOU and iPS-DYP0730 cultures were purchased from the ATCC cell bank (ACS-1023™, ATCC, Manassas, VA, USA). For all pluripotent stem cell passaging, we used a 1 µg/mL dispase solution in DMEM/F12 (Stem Cell Technologies, Vancouver, BC, Canada). For every step, the plastic surfaces were coated with Matrigel solution (1/40 in DMEM/F12) (BD Bioscience, Billerica, MA, USA). The cells were cultured in mTesR1 medium (Stem Cell Technologies, Vancouver, BC, Canada) at 37 °C in a CO_2_-incubator with 5% CO_2_, 5% O_2_ and at 100% humidity. Also, a stable iPS cell line expressing SypHer-2, which had been obtained earlier [25], was used in our work.

The iPS-KYOU, iPS-DYP0730, and iPS-SypHer-2 lines were directed into neural differentiation using the following protocol. Cells were seeded in Ø 6 cm dishes with around 10^6^ cells per dish in mTeSR1 medium. The next day, the medium was changed to Neural Induction Medium (PSC Neural Induction Medium, Life Technologies, Carlsbad, CA, USA). The cells were cultured in this medium for four passages in Ø 6 cm Petri dishes, coated with BD Matrigel (1/40 in DMEM/F12) (BD Bioscience). The resulting neural stem cells (NSCs) were cultured in a Neural Proliferation Medium (NPM, DMEM/F12 (3/1)), with the addition of B27 supplement, bFGF 10 ng/mL, EGF 10 ng/mL, sodium pyruvate, penicillin/streptomycin) for at least three passages. One day before the final neuronal differentiation passage, the neural stem cells were reseeded onto plastic culture dishes and plates coated with BD Matrigel, at a density of 2 × 10^4^ cells/cm^2^. The next day, the medium was changed to N2B27 Medium (DMEM/F12+ Neurobasal Medium (1/1), B27 supplement, N2 Supplement, sodium pyruvate, penicillin/streptomycin), and cultured for a further 28–42 days. The medium was changed every other day.

### 2.2. Obtaining of Spheroids

For spheroid formation the NSC-KYOU, NSC-DYP0730, and NSC-SypHer-2 suspensions were brought to a concentration of 5000 cells/25 µL and dispensed in Petri dishes using the “hanging drops” method. During the day the cells became organized into spherical units. After a day the spheroids were collected and transferred into Petri dishes with ultra-low adhesiveness and cultivated further in suspension form. For imaging analysis, the spheroids were transferred to glass-bottom FluoroDishes (WPI, Sarasota, FL, USA) coated with Matrigel in 0.5 mL DMEM media (Gibco, Thermo Fisher Scientific, Waltham, MA, USA) without phenol red, and allowed to attach for 4–5 h. 

### 2.3. Immunofluorescent Staining

Before staining, the cells were washed with PBS and then fixed for 15 min in 4% paraformaldehyde at room temperature (22 °C–24 °C), then washed three times in PBS (5 min at room temperature), incubated with primary antibodies in blocking solution (PBS with 10% FBS and 0.1% Tryton-X-100) for 1 h at 37 °C, washed again and incubated with the appropriate Alexa Fluor 488/546-conjugated secondary antibodies (1:800, Molecular Probes) for 1 h at 37 °C. After that, the nuclei were counterstained with 1 mg/mL 4′, 6-Diamidino-2-phenylindole dihydrochloride (DAPI). Images were obtained using a CKX41 (Olympus, Tokyo, Japan) fluorescence microscope. Primary antibodies are presented in Table 1.

### 2.4. RT-PCR

Total RNA was isolated using an RNeasy Mini Kit (Qiagen, #74106, Hilden, Germany) with on-column DNA digestion (Qiagen, #79254) according to the manufacturer’s instructions. Reverse transcription was performed using an MMLV RT Kit (Evrogen, #SK021, Moscow, Russia) and oligo(dT)-15 primers. 1 μg of total RNA was added to each reverse transcription reaction. Real-time PCR was performed using qPCRmix-HS-SYBR + Low Rox ready-to-use reaction mixture (Evrogen, #PK156L, Moscow, Russia) and a CFX96 PCR System (Bio-Rad, Berkeley, CA, USA). The PCR protocol was (1) 95 °C for 10 min, (2) 40 cycles of 95 °C for 15 s and 60 °C for 1 min, (3) melt curve analysis with measurements between 60 °C and 95 °C. The primer sequences are given in Table 2. Appropriate genes for normalization factor calculations were chosen using the method described by Vandesompele et al. [30] taking into account our recent experience [31].

### 2.5. Analysis of Spheroids Metabolic Activity by Multiphoton Fluorescence Microscopy and FLIM

To obtain the intensity and fluorescence lifetime images of the intracellular cofactors NAD(P)H and FAD an LSM 880 (Carl Zeiss, Jena, Germany), equipped with a short-pulse femtosecond Ti:Sa laser Mai Tai HP with a pulse repetition rate of 80 MHz, and a duration of 140 ± 20 fsec (Spectra-Physics, Milpitas, CA, USA) and an FLIM system for time resolved microscopy (Becker&Hickle GmbH, Berlin, Germany) were used. Fluorescence of NAD(P)H was excited in two-photon mode at a wavelength of 750 nm. To register fluorescence in the range of 455–500 nm we used an NDD 1512–533 filter (CarlZeiss, Jena, Germany). Flavin fluorescence was excited at a wavelength of 900 nm, while, to detect the signal in the 500–550 nm range, we used an NDD 1512–948 filter (CarlZeiss, Jena, Germany). All studies were conducted under constant conditions in an XL multi S Dark LS incubator (PeСon GmbH, Erbach, Germany) (37 °C and 5% CO_2_). The optical redox ratio was calculated by dividing the corresponding two-photon fluorescence images of the FAD and NAD(P)H using ImageJ 1.52p software (NIH, Bethesda, MD, USA). The FLIM images of NAD(P)H and FAD were processed with SPCImage (Becker & Hickl GmbH, Berlin, Germany). The values of the following parameters were registered: τ1 (ps)—the short fluorescence lifetime that corresponds to the free forms of NAD(P)H and bound forms of FAD; τ2 (ps)—the long fluorescence lifetime that corresponds to the bound forms of NAD(P)H and the free forms of FAD; α1 (%)—the contribution of the short component to the fluorescence lifetime; α2 (%)—the contribution of the long component to the fluorescence lifetime.

### 2.6. Analysis of Spheroids Intracellular pH Using Multiphoton Fluorescence Microscopy

For estimation of the intracellular pH of cells in healthy neuronal spheroids, the genetically encoded sensor SypHer-2 was used. SypHer-2 has two peaks of fluorescence excitation (420 nm and 500 nm), according to the ratio of which the pH value can be calculated (ratiometrically) [32]. In preliminary experiments, calibration had been performed with SypHer-2 to convert conventional pH units to absolute units. For this, calibration solutions were prepared (130 mM potassium glucanate, 20 mM sodium glucanate, 2 mM CaCl_2_, 1 mM MgCl_2_, 30 mM HEPES (pH 6.9–7.9), 30 mM MES (pH 6–6.9), 30 mM Tris (pH 7.9–9.0), 30 mM MOPS (pH 6.9–8.0)) with pH values 6.8, 7.0, 7.2, 7.4, 7.6, 7.8 and 8.0. The spheroids were placed in the calibration solutions with the addition of 10 µM nigericin for 5 min. Next, fluorescence images were obtained and the I488/I405 ratio was calculated. In accordance with the data obtained, calibration curves were plotted for the dependence of the ratio I488/I405 on the pH value. The calibration curves were described by exponential equations. Before imaging, the spheroids were transferred to 35 mm glass bottom dishes (FluoroDish, (WPI, Sarasota, FL, USA)) and FluoroBrite medium (Gibco, Thermo Fisher Scientific, Waltham, MA, USA) without phenol red was added. For the spheroids, the dishes were coated with Matrigel, after which the 3D cultures were left in the medium for 4–5 h for adhesion. Fluorescence images were obtained with an LSM 880 (Carl Zeiss, Jena, Germany). For the genetically encoded pH sensor SypHer-2, fluorescence was excited at wavelengths of 405 nm and 488 nm, followed by detection in the range 500–550 nm. The resulting images were processed using Image J software (NIH, Bethesda, MD, USA) to determine the intensity ratio of the 488 nm to 405 nm excitation (I488/I405). Then, the relative pH values were converted to absolute values using the calibration curve. For spheroids with Down syndrome, staining with a ratiometric pH probe, BCECF (Thermo Fisher Scientific, Waltham, MA, USA), was used. In preliminary experiments, the BCECF had been calibrated to convert pH to absolute units. For staining, a stock solution of the probe in DMSO with a concentration of 1 mM was prepared. Before staining, the spheroids were transferred to 35 mm glass-bottom coated Matrigel plates (FluoroDish, glass thickness 0.17 mm) for 4–5 h for partial adhesion. Then the spheroids were incubated in culture medium containing the probe for 60 min at 37 °C with constant shaking. Fluorescence images were obtained on an LSM 880 (Carl Zeiss, Jena, Germany). For the BCECF, fluorescence was excited at 405 nm and 488 nm, respectively, followed by detection in the 500–550 nm range. The resulting images were processed using Image J software (NIH, Bethesda, MD, USA). 

### 2.7. Statistical Analysis

For energy metabolism research, the images of 9 spheroids of each type and about 90 ROIs were analyzed. For pH analysis, from five to eight spheroids of each type and at least 100 ROI were analyzed. The results were processed and statistically analyzed using EXCEL and the STATISTICA 12 (Tulsa, OK, USA). The redox ratio, FLIM data and I488/I405 ratio were expressed as mean values ± SD. The statistical significance of the differences was confirmed using ANOVA. Specifically, the Bonferroni test was used to determine the group error probability.

## 3. Results

### 3.1. Neural Differentiation in 2D and 3D Conditions, Formation of TS21 and NK Spheroids

We compared the expression of pluripotency markers in the iPS-DYP0730 and iPS-KYOU cell lines using immunochemistry and qRT-PCR. The levels of expression of the pluripotency genes OCT4, SOX2, NANOG and LIN28 [33] and the pluripotency markers DPPA4 and TDGF1 [34] were similar in both cell lines (Figure 1 and Figure 2). Therefore, the iPS-DYP0730 and iPS-KYOU cell lines had retained their pluripotent status.

Next, we performed the differentiation of the iPS-DYP0730 and iPS-KYOU lines into neural spheroids. At the same time, we produced 2D neural cultures from each type of iPSC in order to compare their gene expression profiles during differentiation in 2D and 3D conditions.

We tracked the neural spheroid differentiation using immunochemistry and qRT-PCR. According to the immunochemical analysis, the neural spheroids expressed the TUBB3, GFAP, NESTIN, hNCAM, NeuN and TH proteins (Figure 3). Nestin and hNCAM are markers of neural stem cells and neural progenitors, while GFAP is a glial marker. TUBB3 is expressed in immature post-mitotic neurons, while NeuN and TH are markers of mature neurons. TH is a specific marker of dopamine or forebrain neurons [35].

We also performed qRT-PCR analysis to verify the increasing expression of neural markers and decreasing pluripotent cell markers during the maturation of the spheroids. We compared the levels of expression of GFAP, PAX6, MAPT, SOX2, OCT4 and TDGF1 in the iPSCs, in the 2D neural cell culture and in the 3D spheroids (Figure 4). The gene expression dynamics of the neural markers PAX6, GFAP and MAPT and markers of pluripotency SOX2, OCT4, TDGF1 indicate the successful transition of neural differentiation in both 2D and 3D conditions. We observed a comparatively high level of expression of SOX2 in the 2D neural cultures and spheroids, while OCT4 and TDGF1 expression in these cells was suppressed. Therefore, the iPSCs had been transformed into neural cell types, but some neural progenitors, which express SOX2, still persisted in the sample after differentiation [36]. The large standard deviations in the recorded values of expression of the GFAP and MAPT genes in the 2D neural cells reflect the high degree of variation in the differentiation process for each line.

Next, we compared the levels of expression of genes coding for the proteins responsible for amyloid-β processing (APP, BACE1, BACE2, PSEN1, PSEN2, CD147, TMED10 and DYRK1A, RCAN1, CREB1) associated with Alzheimer’s disease, in the groups of TS21 and NK spheroids, and of the neural differentiation markers GFAP, PAX6 and MAPT (Figure 5). Еxpression of the genes APP, BACE2, RCAN1 DYRK1A, and GFAP was significantly increased. Several genes showed decreased expression in the TS21 spheroids (PAX6, BACE1, PSEN1, PSEN2, and CREB1). Notably, all genes located on chromosome 21 (DYRK1A, APP, BACE2, and RCAN1) showed increased expression levels in the TS21 spheroids.

In addition, we measured the concentrations of two amyloid-β isoforms (Aβ40 and Aβ42) for the TS21 and NK spheroids in conditioned medium. We observed a statistically significant elevation of Aβ40 and Aβ42 secretion in the TS21 spheroids (Figure 6). Aβ40 and Aβ42 secretion in the TS21 spheroids (*n* = 7) was increased by ~95% and ~438% respectively, compared to NK spheroids (*n* = 9). The Aβ42/Aβ40 ratio, also, was ~195% higher in the TS21 spheroids.

### 3.2. Evaluation of TS21 Spheroid Metabolic Activity Using FLIM

To date, many studies have been carried out to analyze the metabolic activity of cells in both two-dimensional and three-dimensional culture models using the fluorescence and FLIM methods [37,38,39]. It has been shown that glycolysis and oxidative phosphorylation are characterized by different fluorescence (optical redox ratio) and FLIM parameters, namely, glycolysis is characterized by a decrease in the redox ratio and an increase in the lifetime contribution of the free form of NAD(P)H (a1), while OXPHOS, by contrast, is represented by an increase in the redox ratio and an increase in the contribution of the bound form of NAD(P)H (a2) [23,24]. 

To study the metabolic status of TS21 cells in spheroids, first we performed analyses of the redox ratios (FAD/NAD(P)H), the fluorescence lifetimes and the fluorescence lifetime contributions of the free and bound forms of NAD(P)H using fluorescence microscopy and FLIM. Evaluation of the redox ratio showed that the values of this parameter were higher in the NK spheroids, compared to TS21 spheroids, where heterogeneity was also observed for this parameter with an increase from the periphery to the center (0.05 ± 0.01 and 0.07 ± 0.01, respectively (*p* = 10^−6^). In general, this indicates that TS21 spheroids are characterized by a more glycolytic phenotype than that found in healthy ones. Statistically significant differences in the mean fluorescence lifetimes were found between the TS21 and NK spheroids. In addition, we demonstrated that 3D model NSCs carrying DS are characterized by lower values of fluorescence lifetime contributions of the bound form of NAD(P)H than in spheroids with the cultivated NK NSCs, indicating their more glycolytic status, which is consistent with the redox ratio data (Figure 7). The redox ratio and FLIM parameters in TS21 and NK spheroids are presented in Table 3.

### 3.3. Evaluation of TS21 Spheroid Intracellular pH Using Fluorescence Microscopy

In preliminary experiments, calibrations were carried out with the pH-sensors SypHer-2 and BCECF to convert conventional pH units to absolute units (Figure 8a,b). In the NK spheroids, the average value of the fluorescence intensity ratio (I488/I405) was 0.281± 0.003, which, according to the calibration curve, corresponded to absolute values of pH of 7.05 ± 0.05. (Figure 8c). In the TS21 spheroids, the mean value of the fluorescence intensity ratio was 1.52 ± 0.08, which, according to the calibration curve, corresponded to absolute pH values of 7.18 ± 0.06 (Figure 8d). Thus, we have demonstrated a statistically significant difference between the intracellular pH of TS21 spheroids and NK spheroids (*p* = 0.00143). These results correlate well with the data on the metabolic status of cells in neuronal 3D models. Namely, the more alkaline values of intracellular pH in TS21 spheroids correlate with the predominance of glycolysis, while cells in healthy spheroids are characterized by more acidic pH values, where OXPHOS prevails.

## 4. Discussion

3D spheroids and organoids from donor iPSCs with hereditary diseases are promising objects for studying features of the pathological processes, and for helping in the selection of drugs and tracking the mechanisms of their action. For instance, it has been shown that iPSCs from donors with hereditary neurodegenerative diseases have the ability to reproduce the pathological processes in vitro [40]. Hereditary diseases associated with genetic abnormalities, such as DS, directly affect the cells’ metabolism, the balance of which, in turn, is directly related to the intracellular pH. Therefore, the changes in the metabolism can serve as potential parameters for the study of neurodegenerative disease phenotypes and the therapeutic effects of potential drugs. 

In our study, we obtained neural spheroids made from a Down syndrome donor’s iPSCs. Using several independent techniques of protein and gene expression analysis we showed that NSCs differentiated from such iPSCs carrying DS can successfully form 3D spheroids. We demonstrated for the first time the differences in cellular metabolism and intracellular pH between NK neural spheroids and TS21 spheroids, using both fluorescent and FLIM methods. The obtained data indicate the predominance of glycolysis in TS21 spheroids compared with NK spheroids. In addition, assessment of the intracellular pH in spheroids with and without pathology, using fluorescence probes (BCECF, SypHer-2) and fluorescence microscopy showed a more alkaline intracellular pH of the cytoplasm in TS21 spheroids compared to NK ones.

To date, many different protocols for deriving specific neural organoids and spheroids have been developed. Thus, organoids for modeling the central nervous system have been obtained, including those representing specific brain compartments (forebrain [41,42], hypothalamus [43], cerebellum [44], midbrain [43,45,46], cerebral cortex [47,48,49] and retinal organoids [50]. 

The use of 3D organoids and spheroids makes in vitro studies more equivalent to in vivo conditions [51,52]. We have developed an original and effective protocol for 3D spheroid generation. We were able to observe the expression of markers characteristic of mature differentiated neural, glial and neural precursor cells (SOX2, PAX6, MAPT genes, and the Nestin, hNCAM, TUBB3, TH, NeuN and GFAP proteins) [35,36]. Spheroids possess many of the hallmarks of DS, particularly abnormal amyloid-β secretion.

Maintaining the functional state of energy metabolism in accordance with the physiological needs of cells is a necessity for their normal functioning. ATP in cells is known to be synthesized through the two metabolic pathways, glycolysis and oxidative phosphorylation (OXPHOS). It is known that stem cells, including neuronal stem cells (NSCs), rely primarily on glycolysis, while differentiated cells, like neurons, use mainly OXPHOS [53,54,55]. The predominance of glycolysis in NSCs is associated with their presence in niches, where the oxygen content is lower than in the surrounding tissues [53]. However, a change in the energy metabolism of differentiated cells may indicate the course of a pathological process. Many neurodegenerative diseases, such as Alzheimer’s disease [56], Parkinson’s disease [57], Huntington’s disease [58], Amyotrophic lateral sclerosis [59] and Down syndrome [11] are characterized by impaired energy metabolism in the cells. In Down syndrome, several genes linked to 21 chromosomes are known to repress mitochondrial biogenesis [11]. Deterioration of mitochondrial functioning leads to disruption of the processes of OXPHOS and a shift in ATP production to aerobic glycolysis [60]. OXPHOS is known to be the main supplier of ATP for mature neurons [10]. When OXPHOS is impaired, the nervous system is primarily affected [61]. Therefore, an urgent task is to find an effective treatment aimed at correcting the metabolic shift in neurodegenerative diseases. One of the known ways of correcting metabolic disorders during neurodegenerative diseases is by increasing glucose uptake and its dissimilation, which can be achieved by pharmacological and genetic manipulations [62,63,64]. 

Early research has shown that changes in cellular metabolism provide a clear differentiating parameter by which healthy neuronal cells can be distinguished from cells with neurodegenerative diseases. For these purposes, the values of the FAD/NAD(P)H redox ratio, as well as the fluorescence lifetimes and the contributions of the bound form NAD(P)H, were measured and evaluated. Estimation of the redox ratio showed that the value of this parameter is two times lower in DS spheroids compared to healthy ones. Analysis of FLIM parameters revealed that the DS spheroids are also characterized by their lower fraction of the bound form of NAD(P)H than in healthy spheroids and demonstrated a greater bias toward glycolytic status. Generally, the fluorescence and FLIM characteristics of cells in DS spheroids confirmed the reduction of OXPHOS, which has been associated with genetic abnormalities. To the best of our knowledge, such changes of cell metabolism in 3D models carrying Down syndrome have never previously been measured using fluorescence microscopy and FLIM. However, it should be noted that FLIM is already used for the in vitro analysis of metabolic changes in 2D cell cultures during the development of neurodegenerative diseases, such as Parkinson’s disease [65] and Huntington’s disease [66]. In studies conducted on 3D models created from stem cells, FLIM have also been used for analysis of the cell cycle [67], stem cell niches [68] and cell metabolism [20,69].

Such a change in the metabolic activity of cells is directly related to a change in the pH. Therefore, the level of intracellular pH can also be used as a parameter, the shift of which can indicate the presence of a pathological process. In our work, intracellular pH in spheroids with and without DS was measured. For this aim, the fluorescence intensities of the fluorescent probe—BCECF and the genetically-encoded sensor—SypHer-2 upon excitation at two wavelengths were measured, followed by calculations of the ratios of I488 to I405. According to the calibration curves, it was shown that spheroids carrying DS have more alkaline pH values than healthy ones (pH = 7.18 and 7.1, respectively). It is known that the intracellular pH of healthy cells is maintained within a narrow interval 7.0–7.2 [29]. Rodeau and co-workers demonstrated that the intracellular pH of spinal cord neurons cultured from rat embryos was 7.18 ± 0.03 [70]. Despite the fact that we have shown a statistically significant difference between the pH values in healthy neuronal spheroids and spheroids with DS, both values were in the typical physiological range. Therefore, in our case, pH cannot be seen as an independent differentiating parameter, but it does serve as an additional one supporting analyses of metabolism. It is also known that a more alkaline intracellular pH favors glycolysis by increasing the activity of glycolytic enzymes such as lactate dehydrogenase and phosphofructokinase 1 [71]. In our case, the pH data correlate well with the data on metabolic status, namely, TS21 spheroids have more alkaline pH values and characterize more glycolytic status. However no data were provided about changes in the intracellular pH in neurodegenerative diseases, although it is known that a reverse pH gradient is characteristic of cancer cells. This is due to an increase in the number of membrane ion transporters, which leads to the active transfer of protons out of the cell and to acidification of the extracellular environment [72,73,74]. In neurobiology, fluorescent pH probes such as BCECF have usually been used for in vitro and ex vivo examination of pH regulation by cells from different areas of the brain [72,75]. In earlier studies, the fluorescent, genetically-encoded pH sensor SypHer-2 was used to analyze intracellular pH in tumor spheroids [76,77]. 

Thus, we have shown the capabilities of fluorescence microscopy and FLIM for assessing the metabolism and pH of cells in 3D models of neurodegenerative diseases. Fluorescence microscopy and FLIM can be used to quickly assess the disease phenotype and the effectiveness of therapeutic effects on cells for the correction of metabolic disorders. Moreover, these optical techniques may be effectively combined with immunocytochemical, histochemical, colorimetric, biochemical, and genetic methods for the full screening of 3D models of neurodegenerative diseases.

## 5. Conclusions

In our study, a comprehensive analysis of the metabolic status and pH levels in NK neural spheroids and TS21 spheroids was carried out using fluorescence microscopy and FLIM. We have shown, for the first time, that glycolysis is the dominant way of providing energy in TS21 spheroids compared to NK spheroids, where OXPHOS prevails. In addition, using the pH-sensitive sensors BCECF and SypHer-2 and fluorescence microscopy, it was shown that cells in TS21 spheroids are characterized by a more alkaline pH compared to NK neural spheroids. Metabolic imaging methods in combination with endogenous markers are promising tools for analyses of the energy metabolism of three-dimensional models, based on patient-specific cells, which can be used to help analyze the disease phenotype and the effectiveness of metabolic treatments of pathological conditions such as Down syndrome.

## Figures and Tables

**Figure 1 biomedicines-09-01741-f001:**
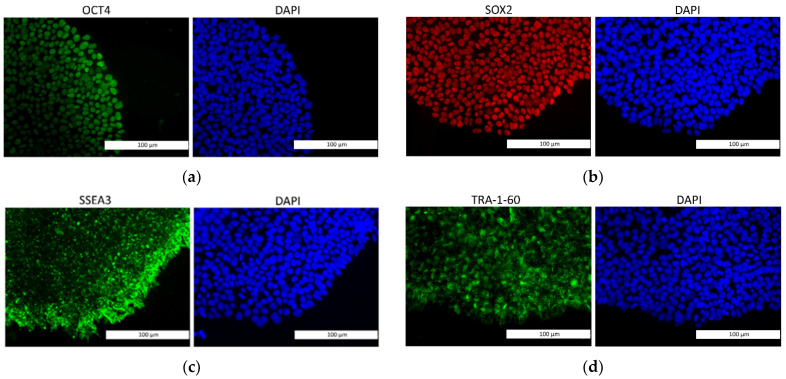
Immunohistochemical staining for evaluation of the expression of pluripotency markers: (**a**) OCT4; (**b**) SOX2; (**c**) SSEA3; (**d**) TRA-1-60. Expression of OCT4, SOX2, SSEA3, TRA-1-60 was found in both karyotypes. Fluorescence microscopy, the scale length in all pictures is 100 µm.

**Figure 2 biomedicines-09-01741-f002:**
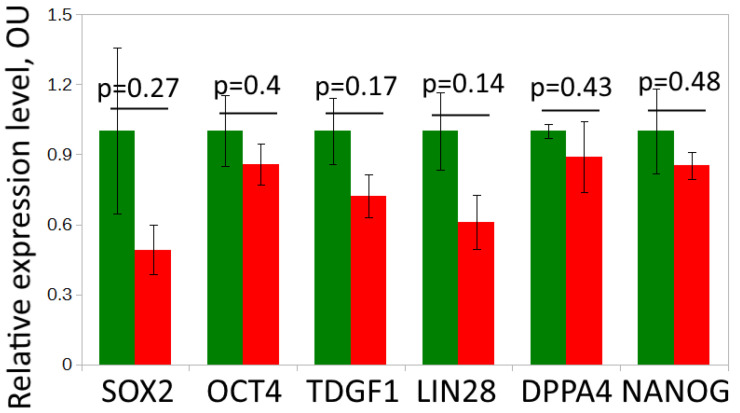
Analysis of the mRNA expression levels of OCT4, SOX2, NANOG, LIN28, DPPA4, and TDGF1 in lines iPS-KYOU (green) and iPS-DYP (red). Two biological repeats were performed. Housekeeping genes C1ORF43 and REEP5 were used for the normalization of the qRT-PCR data.

**Figure 3 biomedicines-09-01741-f003:**
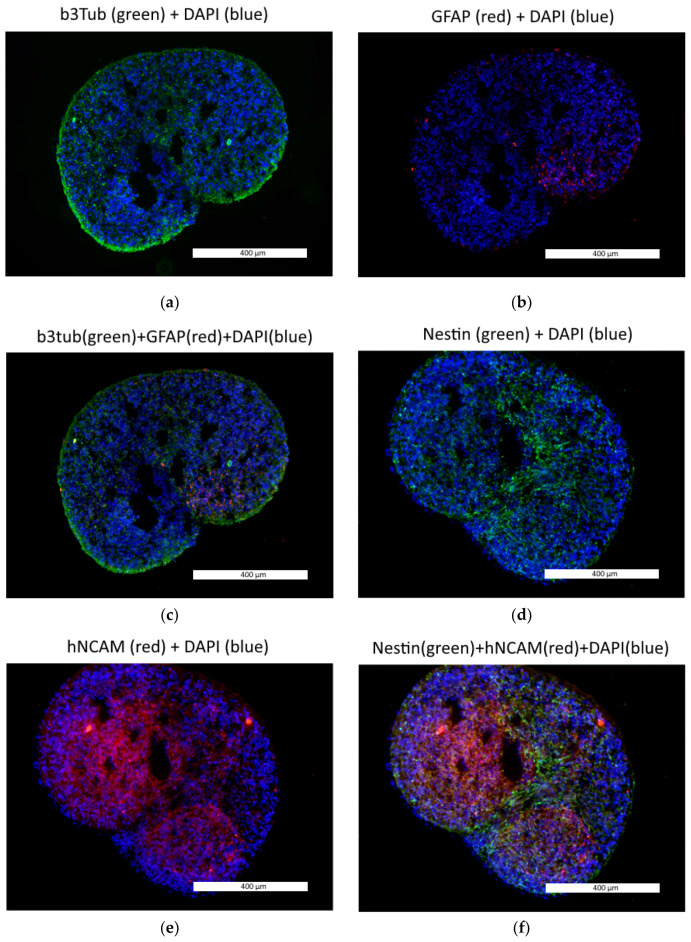
The 3D neural spheroids expressed neural cell markers: (**a**) TUBB3 (green); (**b**) GFAP (red); (**c**) TUBB3 (green) and GFAP (red); (**d**) Nestin (green); (**e**) hNCAM (red); (**f**) Nestin (green) and hNCAM (red); (**g**) NeuN (red); (**h**) TH (red); DAPY-stained cell nuclei in all pictures are blue. Fluorescence microscopy, the scale length in all pictures is 400 µm.

**Figure 4 biomedicines-09-01741-f004:**
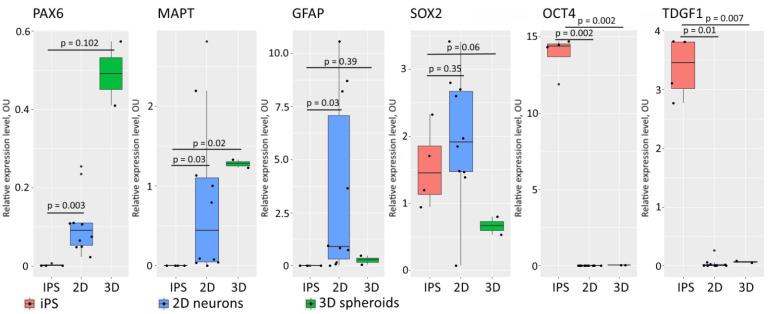
Tracking of changes in expression of pluripotency markers and markers of neural cells during differentiation. Pluripotency markers diminished, while neural markers increased during differentiation. Housekeeping genes REEP5, GAPDH and HMBS were used for normalization of the qRT-PCR data.

**Figure 5 biomedicines-09-01741-f005:**
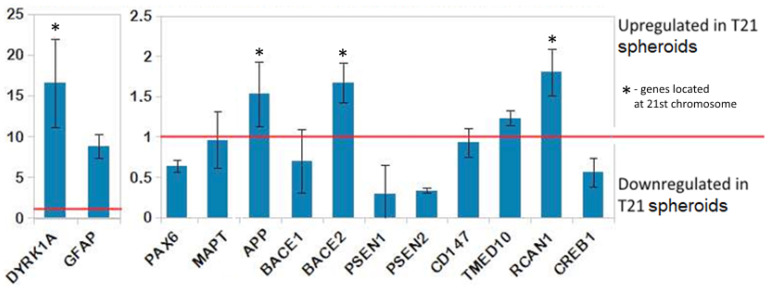
Expression level ratio between TS21 and NK spheroids. The red line marks equal expression in the TS21 and normal karyotype spheroids. Ratio values greater than 1 indicate upregulated genes in the TS21 spheroids. Ratio values less than 1 indicate downregulated genes in the TS21 spheroids respective to the normal karyotype. Asterisks (*) label genes carried on chromosome 21. The normalization factors for qRT-PCR analysis were calculated based on REEP5, C1orf43, and HMBS.

**Figure 6 biomedicines-09-01741-f006:**
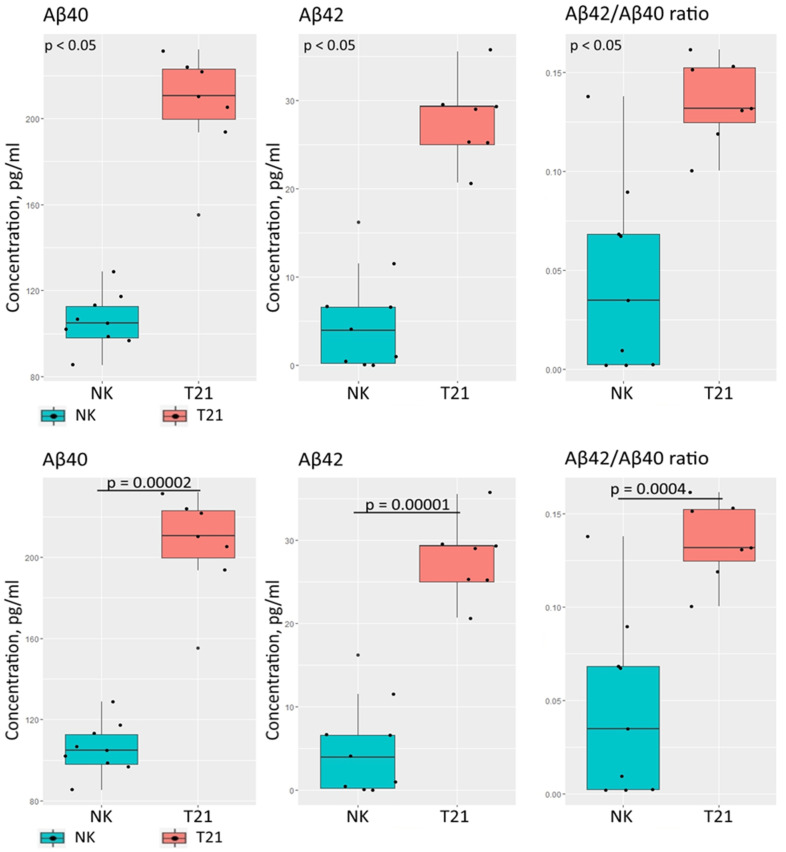
Immunofluorecent quantification of Aβ40 and Aβ42 in conditioned medium. TS21 spheroids secrete more Aβ40 and Aβ42. The proportion of Aβ42 is higher in TS21 spheroids. The differences are statistically significant according to the *T*-test (for NK spheroids *n* = 9, for TS21 spheroids *n* = 7).

**Figure 7 biomedicines-09-01741-f007:**
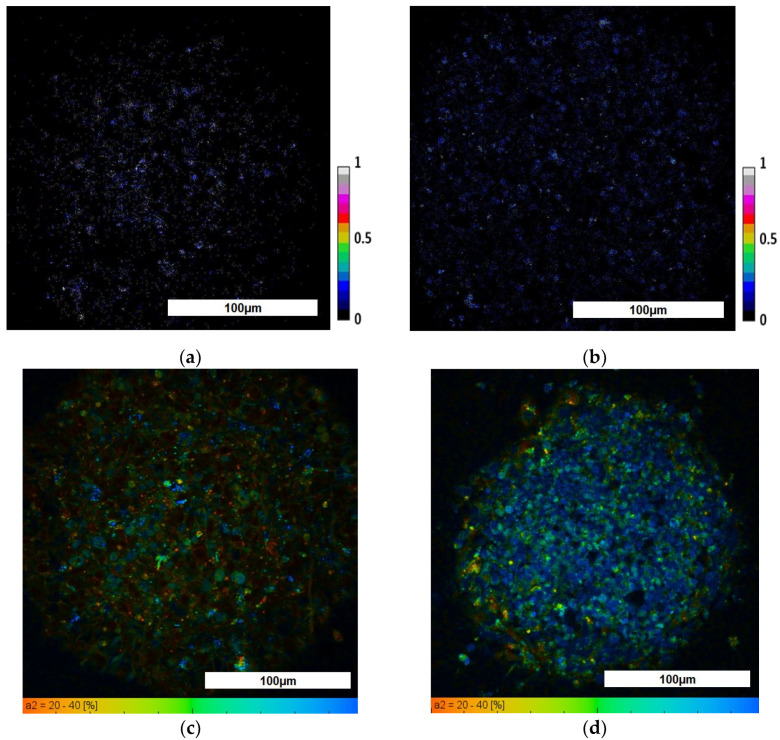
Multiphoton fluorescence microscopy and FLIM of NAD(P)H in TS21 and NK spheroids: (**a**) Optical redox image (fluorescence intensity of FAD/NAD(P)H) of TS21 spheroid; (**b**) Optical redox image (fluorescence intensity of FAD/NAD(P)H) of NK spheroid; (**c**) Pseudocolor—coded image of the protein-bound form of NAD(P)H (α2, %) of TS21 spheroid; (**d**) Pseudocolor—coded image of the protein-bound form of NAD(P)H (α2, %) of NK spheroid. For NAD(P)H: excitation—750 nm, detection—455–500 nm; for FAD: excitation—900 nm, detection—500–550 nm. Field of view 213 × 213 μm (512 × 512 pixels). Scale bar: 100 µm.

**Figure 8 biomedicines-09-01741-f008:**
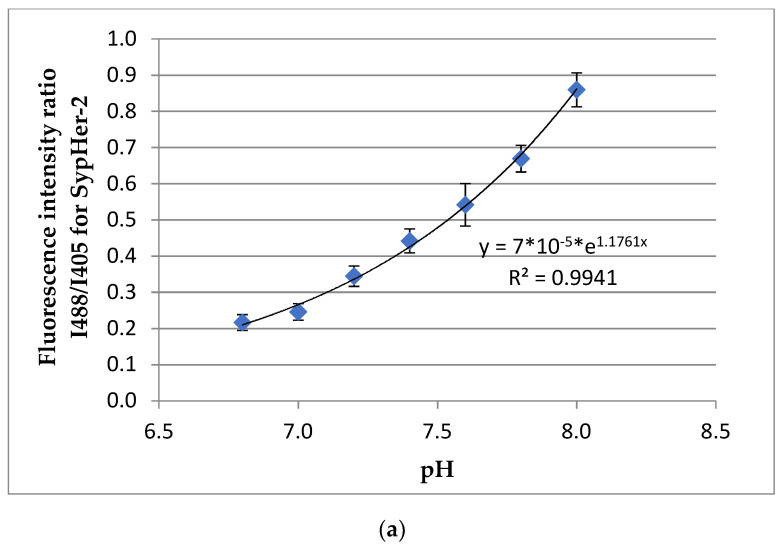
Analysis of the intracellular pH in TS21 and NK spheroids: (**a**) Calibration curve of the dependence of the ratio of fluorescence intensities on the pH for SypHer-2 in NK spheroids (Mean ± SD); (**b**) Calibration curve of the dependence of the ratio of fluorescence intensities on the pH for BCECF in TS21 spheroids (Mean ± SD); (**c**) Image of the ratio of the fluorescence intensity for 488 nm to 405 nm excitation in NK spheroids-SypHer-2. Excitation of SypHer-2 fluorescence at 488 and 405 nm, fluorescence detection at 500–550 nm. Image size 213 × 213 μm; (**d**) Image of the ratio of fluorescence intensity for 488 nm to 405 nm excitation in TS21 spheroids stained with BCECF. Excitation of BCECF fluorescence at 488 and 405 nm, fluorescence detection at 500–550 nm. Scale bar: 100 µm.

**Table 1 biomedicines-09-01741-t001:** Primary antibodies for immunofluorescent staining.

Antibodies Names	Producer (cat. №)
Anti-Oct4	Millipore (MAB4401)
Ati-Sox2	Cell Signaling Technology (3579)
Anti-SSEA3	Abcam (ab16286)
Anti-TRA-1-60	Abcam (ab16288)
Anti-TUBB3	Millipore (MAB1637)
Anti-GFAP	Millipore (04-1031)
Anti-Nestin	Millipore (MAB5326)
Anti-hNCAM	Abcam (ab75813)
Anti-NeuN	Abcam (ab104225)
Anti-TH	Abcam (ab75875)

**Table 2 biomedicines-09-01741-t002:** The primer sequences for RT-PCR.

Primer Target	Primer Sequence (5′→3′)
SOX2	Forward Primer-TGCGAGCGCTGCACAT
	Reverse Primer-GCAGCGTGTACTTATCCTTCTTCA
OCT4	Forward Primer-ACCCACACTGCAGCAGATCA
	Reverse Primer-CACACTCGGACCACATCCTTCT
Nanog	Forward Primer-ACAACTGGCCGAAGAATAGCA
	Reverse Primer-GGTTCCCAGTCGGGTTCAC
Lin28	Forward Primer-AGTCAGCCAAGGGTCTGGAAT
	Reverse Primer-CCGCCTCTCACTCCCAATA
DPPA4	Forward Primer-GACCTCCACAGAGAAGTCGAG
	Reverse Primer-TGCCTTTTTCTTAGGGCAGAG
TDGF1	Forward Primer-TTTGAACTGGGATTAGTTGCCG
	Reverse Primer-GGGGCCAAATGCTGTCATCT
GFAP	Forward Primer-AGGTCCATGTGGAGCTTGAC
	Reverse Primer-GCCATTGCCTCATACTGCGT
MAPT	Forward Primer-TTTGGTGGTGGTTAGAGATATGC
	Reverse Primer-CCGAGGTGCGTGAAGAAATG
PAX6	Forward Primer-AGTGCCCGTCCATCTTTGC
	Reverse Primer-CGCTTGGTATGTTATCGTTGGT
DYRK1A	Forward Primer-AAGAAGCGAAGACACCAACAG
	Reverse Primer-TTTCGTAACGATCCATCCACTTT
APP	Forward Primer-CAAGCAGTGCAAGACCCATC
	Reverse Primer-AGAAGGGCATCACTTACAAACTC
BACE1	Forward Primer-TCTGTCGGAGGGAGCATGAT
	Reverse Primer-GCAAACGAAGGTTGGTGGT
BACE2	Forward Primer-TGCCTGGGATTAAATGGAATGG
	Reverse Primer-CAGGGAGTCGAAGAAGGTCTC
PSEN1	Forward Primer-ACAGGTGCTATAAGGTCATCCA
	Reverse Primer-CAGATCAGGAGTGCAACAGTAAT
PSEN2	Forward Primer-TCACTCTGTGCATGATCGTGG
	Reverse Primer-GTGAATGGCGTGTAGATGAGC
CD147	Forward Primer-CTCCCAGAGTGAAGGCTGTG
	Reverse Primer-ACTCTGACTTGCAGACCAGC
TMED10	Forward Primer-TGGAGGCGAAAAATTACGAAGA
	Reverse Primer-CTAGGCGTCGCAGCTCTAC
RCAN1	Forward Primer-GCGTGGTGGTCCATGTATGT
	Reverse Primer-TGAGGTGGATCGGCGTGTA
CREB1	Forward Primer-CCACTGTAACGGTGCCAACT
	Reverse Primer-GCTGCATTGGTCATGGTTAATGT

**Table 3 biomedicines-09-01741-t003:** Data on the redox ratio and FLIM parameters in TS21 and NK spheroids (mean ± SD, *p* = 10^−6^).

Type of Spheroids		Redox Ratio	τm (ps)	τ1 (ps)	τ2 (ps)	α2, %
NK		0.13 ± 0.02	1301.21 ± 54.06	462.74 ± 31.76	2373.12 ± 67.03	31.33 ± 1.42
TS21	periphery	0.05 ± 0.01	939.39 ± 65.16	407.72 ± 17.16	2528.2 ± 77.14	27.18 ± 1.12
	center	0.07 ± 0.01				

## Data Availability

Not applicable.

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
