# Peer review of "Energy Metabolism and Intracellular pH Alteration in Neural Spheroids Carrying Down Syndrome"

_biomedicines, 2021, doi:10.3390/biomedicines9111741_

Round 1
Reviewer 1 Report
The authors did an excellent research and presented their study nicely. However, the data presentation should be more clear. Please see the attachment for details.
The authors should check their manuscript for similar mistakes.

Reviewer 2 Report
In the manuscript „Energy metabolism and intracellular pH alteration in neural spheroids carrying Down syndrome“, its authors, Kashirina A, Gavrina A et al. describe changes in energy metabolism found in spheroids grown from iPS cells, obtained from 21 chromosome trisomic fibroblasts, in comparison with their normal counterparts. Cells were differentiated in vitro and grown as 3D spheroids. Expression of a set of genes, involved in signaling, as well as differentiation and stem cell phenotype markers were assessed by qRTPCR. Differences in beta amyloid extracellular ratios and related gene expression were shown in cells derived from DS iPS cells. Intracellular pH was determined by multiphoton fluorescence microscopy and FLIM of NADH in spheroids and it was found that DS derived spheroids have higher intracellular pH, possibly affecting cellular metabolism.
The manuscript is well written. In Introduction there is sufficent data to explain the topic. Materials and methods are detailed. Results are also well presented, as well as discussion.
Minor comments
Lines 67-69: sentence reconstruction (without such as)
Lines 84, 85, 95, 97, 225, 428: pH
Line 114: iPS
Line 117: in dishes
Line 118, 126: power in superscript
Line 121: sentence without then.
Lines 123, 258: units should be separated from numbers
Line 204: transferred to dishes
Line 224: sentence reconstruction
Line 257: DAPI
Lines 315, 330: p value can be presented as 10-x to avoid 0,0000
Line 350: why there is no control cells stained with BCEFC
Line 439, 458: predominance of glycolysis would mean that glycolysis is the dominant way of providing energy in DS cells, in comparison with OXPHOS
